# Alterations of the Platelet Proteome in Lung Cancer: Accelerated F13A1 and ER Processing as New Actors in Hypercoagulability

**DOI:** 10.3390/cancers13092260

**Published:** 2021-05-08

**Authors:** Huriye Ercan, Lisa-Marie Mauracher, Ella Grilz, Lena Hell, Roland Hellinger, Johannes A. Schmid, Florian Moik, Cihan Ay, Ingrid Pabinger, Maria Zellner

**Affiliations:** 1Clinical Division of Haematology and Haemostaseology, Department of Medicine I, Medical University of Vienna, 1090 Vienna, Austria; huriye.ercan@meduniwien.ac.at (H.E.); lisa_marie_mauracher@gmx.at (L.-M.M.); ella.grilz@gmx.net (E.G.); lena.hell@meduniwien.ac.at (L.H.); florian.moik@meduniwien.ac.at (F.M.); cihan.ay@meduniwien.ac.at (C.A.); 2Centre for Physiology and Pharmacology, Institute of Vascular Biology and Thrombosis Research, Medical University of Vienna, 1090 Vienna, Austria; johannes.schmid@meduniwien.ac.at; 3Department of Anaesthesiology and Intensive Care, Danube Hospital, 1220 Vienna, Austria; 4Centre for Physiology and Pharmacology, Institute of Pharmacology, Medical University of Vienna, 1090 Vienna, Austria; roland.hellinger@meduniwien.ac.at; 5I. M. Sechenov First Moscow State Medical University, 119991 Moscow, Russia

**Keywords:** cancer, lung cancer, thrombosis, platelets, proteomics, coagulation factor XIII, unfolded protein response, protein disulfide-isomerase, endoplasmic reticulum chaperones, therapeutic target

## Abstract

**Simple Summary:**

The risk of venous thromboembolism in cancer is nine times higher than in the general population and the second leading cause of death in these patients. Tissue factor and downstream plasmatic coagulation cascade are largely responsible for the risk of thrombosis in cancer. In recent years, it has been increasingly recognised that platelets also play a central role in tumour growth and cancer-associated thrombosis. The underlying molecular mechanisms are largely unknown. In order to comprehensively investigate the biochemical changes in platelets from cancers with high risk of thrombosis, we examined the platelet proteome of brain and lung cancer patients in comparison to sex and age-matched healthy controls. However, we only found alterations in lung cancer, where some of these platelet proteins directly promote thrombosis. One example is the increased amount of the enzyme protein disulfide isomerase, which is clinically investigated as an antithrombotic drug target of the plant-based flavonol quercetin.

**Abstract:**

In order to comprehensively expose cancer-related biochemical changes, we compared the platelet proteome of two types of cancer with a high risk of thrombosis (22 patients with brain cancer, 19 with lung cancer) to 41 matched healthy controls using unbiased two-dimensional differential in-gel electrophoresis. The examined platelet proteome was unchanged in patients with brain cancer, but considerably affected in lung cancer with 15 significantly altered proteins. Amongst these, the endoplasmic reticulum (ER) proteins calreticulin (CALR), endoplasmic reticulum chaperone BiP (HSPA5) and protein disulfide-isomerase (P4HB) were significantly elevated. Accelerated conversion of the fibrin stabilising factor XIII was detected in platelets of patients with lung cancer by elevated levels of a coagulation factor XIII (F13A1) 55 kDa fragment. A significant correlation of this F13A1 cleavage product with plasma levels of the plasmin–α-2-antiplasmin complex and D-dimer suggests its enhanced degradation by the fibrinolytic system. Protein association network analysis showed that lung cancer-related proteins were involved in platelet degranulation and upregulated ER protein processing. As a possible outcome, plasma FVIII, an immediate end product for ER-mediated glycosylation, correlated significantly with the ER-executing chaperones CALR and HSPA5. These new data on the differential behaviour of platelets in various cancers revealed F13A1 and ER chaperones as potential novel diagnostic and therapeutic targets in lung cancer patients.

## 1. Introduction

Patients with cancer have an increased risk to develop venous thromboembolism (VTE) [1]. This complication occurs in up to 20% of affected individuals per year [2] and is associated with high morbidity and mortality [3,4]. Malignancies with the highest incidence of VTE are brain and lung cancer [5]. Platelets are thought to play an important role in the prothrombotic state and the development of cancer-associated VTE as well as arterial thromboembolism [6]. Besides their crucial functions in primary haemostasis, platelets are also important players in the progression of VTE. Various platelet-associated parameters were already identified to be linked with increased risk of VTE in patients with cancer, such as high platelet counts [7] and exposed phosphatidylserine on platelets [8]. High plasma levels of soluble P-selectin (sCD62P) indicate cancer-associated platelet activation [9]. In a previous study of patients with different types of cancer, we found a statistical association of increased VTE risk and poor survival rate with lower surface expression of receptors important for platelet activation, namely CD62P and glycoprotein IIb/IIIa, a complex of integrin alpha IIb (ITGA2B) and beta 3 (ITGB3) [5]. The decline in these platelet receptors is supposed to be caused by shedding during continuous platelet activation, more strongly ongoing in cancer patients with increased mortality and VTE rate [5]. Platelet activation is also suggested to play a role in the progression of cancer and metastasis formation, as shown by experimental studies in vitro [10] and in knock-out mice with depleted peripheral platelets [11]. Furthermore, in clinical studies anti-platelet therapy with aspirin reduced the risk of cancer and metastasis development [12,13]. Although many studies have shown connections between platelets, cancer and thrombosis, the molecular mechanisms are not entirely understood. Despite this recognised importance of platelets in cancer [14,15] and in many other prothrombotic conditions, clinical studies with comprehensive and unbiased analysis of the platelet proteome, in particular from well-defined high thrombosis risk patient cohorts, have scarcely been performed. In a recent study, we investigated the platelet proteome of Lupus anticoagulant (LA)-positive patients, who are at high risk to develop thrombosis. These investigations revealed changes of functional important platelet proteins, which uncovered new prothrombotic triggers of primary haemostasis [16].

Due to the lack of a nucleus, platelets have a rather small and rapidly declining amount of mRNA and in line with that also a low rate of protein synthesis. Consequently, regulatory mechanisms are mostly detectable on the level of pre-existing proteins and their post-translational modifications (PTMs). A proteomic investigation of these processes might help to elucidate the development of cancer-associated VTE and may contribute substantially to better diagnostics, prevention and therapy, and thus ultimately to patient survival.

Today, “bottom-up” proteome technologies are state-of-the-art and support the analysis of thousands of proteins with automated LC-MS systems, which enables very high sample throughputs to be achieved [17]. Here, liquid-handling workstations are a further important development of the robotized preparation of clinical samples for these technologies of proteomics analysis [18]. For the simultaneous qualitative and quantitative characterisation of intact proteins and their PTMs (= proteoforms), however, the older gel-based “top-down” proteomics technology is still the most practical method [19,20]. This gel-based method is just much more labour intensive as it cannot be automated. However, since we also expect PTM-based changes in the haemostasis-regulating platelets, the fluorescence-based “top-down” two-dimensional differential gel electrophoresis (2D-DIGE) was selected to reveal, in one analytical process, the qualitative and quantitative cancer-related differences of proteins and corresponding PTMs [21]. Protein spots of interest were worked up for proteomics analysis by mass spectrometry (MS) to obtain protein identities.

Despite the important role of platelets in the development of cancer-associated VTE, no prospective systemic study investigating the respective platelet proteome has been performed so far. To address this, we investigated the platelet proteome in patients with two different types of cancer to identify potential mechanisms of increased risk for VTE and/or mortality and possible molecular patterns that are specific for the cancer type. Our data demonstrate that several proteins and/or corresponding proteoforms were significantly changed in the platelet proteome of patients with lung cancer, but no changes in the platelet proteome were detectable in patients with brain tumours.

## 2. Materials and Methods

### 2.1. Study Design

Forty-one patients with cancer (lung and brain) enrolled in the framework of the Vienna Cancer and Thrombosis Study (CATS) were included in this analysis. CATS is a prospective, observational, single-centre cohort study started in October 2003 at the Medical University of Vienna. Patients included in the present study were recruited between 2014 and 2017. All detailed inclusion and exclusion criteria were previously described [22,23]. Briefly, adult patients (≥18 years) with newly diagnosed cancer or cancer progression after partial or complete remission were enrolled in the study. Patients who were not treated with anticoagulant drugs in the past three months, did not undergo chemotherapy in the last 3 months or receive radiotherapy or surgery within the past 2 weeks, were informed about the study design and gave written consent. At study inclusion, patient history was documented with a structured interview and furthermore, patients enrolled in the CATS study were monitored with a structured questionnaire on a three-month basis for occurrence of VTE and death. For this study and its hypothesis generating approach we used a case control design and included a matched healthy control cohort. Follow ups were made after 3 and 6 months and two years.

Forty-one age- and sex-matched healthy volunteers were included after giving written informed consent. Venous blood was drawn from cancer patients and healthy controls for platelet proteome investigations and routine and coagulation parameter analysis as well as for further experimental approaches. Data on the study populations can be found in Table 1. This study was approved by the Ethics Committee of the Medical University of Vienna in accordance with the Declaration of Helsinki (EC no. 126/2003 and 039/2006).

### 2.2. Blood Collection, Washed Platelet and Plasma Isolation

For platelet isolation, blood was drawn from an antecubital vein into 3.5 mL vacuum tubes containing CTAD (0.129 mM trisodium citrate, 15 mM theophylline, 3.7 mM adenosine, 0.198 mM dipyridamole; Greiner Bio-One, Kremsmünster, Austria) as anticoagulant. The first tube drawn was discarded to avoid any contaminations. Immediately following blood draw, CTAD tubes was centrifuged at 120× *g* for 20 min without brake at room temperature. Obtained platelet-rich plasma (PRP) was carefully transferred into a fresh tube containing prostacyclin I_2_ (0.8 µM, PGI_2_; Sigma-Aldrich, St. Louis, MO, USA) to prevent platelet aggregation and degranulation during the following washing steps. PRP was pelleted by centrifugation for 3 min at 3000× *g* at room temperature and the protein pellets were washed twice in phosphate-buffered saline (w/o: Ca^2+^ and Mg^2+^) containing PGI_2_ (0.8 µM). Before the last centrifugation step, platelet count was determined with a Sysmex XN-350 haematocytometer (Sysmex, Kobe, Japan). After the last centrifugation step the supernatant was completely taken off and the platelet pellet was snap-frozen in liquid nitrogen and stored at −80 °C until processing. For preparation of plasma, blood was drawn into 3.5 mL vacuum tubes containing sodium citrate (0.129 mM citrate; Greiner Bio-One, Kremsmünster, Austria). The blood was centrifuged at 2500× *g* for 15 min at 15 °C to separate the cellular fraction, and the plasma supernatant was stored at −80 °C.

### 2.3. Platelet Preparation for Two-Dimensional Fluorescence Differential Gel Electrophoresis (2D-DIGE) Analysis

The frozen platelet protein pellets were resolubilized in urea-sample buffer (7 M urea, 2 M thiourea, 4% CHAPS, 20 mM Tris-HCl pH 8.68) and incubated for 2 h at 4 °C under agitation (800 rpm). Protein quantitation of individual samples was done in triplicate with a Coomassie brilliant blue protein assay kit (Pierce, Thermo Scientific, Rockford, IL, USA). The internal standard (IS) was made by pooling the same protein amounts from each platelet sample of all included study participants. Platelet protein samples and IS were aliquoted and stored at −80 °C.

### 2.4. Platelet Proteome Analysis by 2D-DIGE

Prior to electrophoresis, proteins were labelled with fluorescent cyanine dyes (5 pmol of CyDyes per µg of protein; GE Healthcare, Uppsala, Sweden) according to previous concentrations [16]. The IS was always labelled using Cy2, while Cy3 and Cy5 were used alternately for study samples. Briefly, IPG-Dry-Strips (24 cm, pH 4–7, GE Healthcare, Uppsala, Sweden) were rehydrated for 11 h with 450 µL rehydration buffer (7 M urea, 2 M thiourea, 70 mM DTT, 0.5% pH 4–7 ampholyte; Serva, Heidelberg, Germany) mixed with a total of 36 µg (2 × 12 µg sample + 1 × 12 µg IS) of alternatively Cy-labelled sample. Isoelectric focusing (IEF) was performed on a Protean I12 IEF unit (Bio-Rad) until 30 kVh was reached.

After IEF, the strips were first equilibrated with gentle shaking in 12.5 mL of equilibration buffer 1 (1% DTT, 50 mM Tris-HCl pH 8.68, 6 M Urea, 30% glycerol and 2% SDS) for 20 min followed by incubation in equilibration buffer 2 (2.5% iodoacetamide, 50 mM Tris-HCl pH 8.68, 6 M Urea, 30% glycerol and 2% SDS) for 15 min. Each of the IPG strips was transferred onto 11.5% acrylamide gel (26 × 20 cm, 1 mm gel) and sealed with low-melting agarose sealing solution (375 mM Tris-HCl pH 8.68, 1% SDS, 0.5% agarose). The SDS-PAGE was performed using an Ettan DALTsix electrophoresis chamber (GE Healthcare, Uppsala, Sweden) under the following conditions: 35 V for 1 h, 50 V for 1.5 h and finally 110 V for 16.5 h at 10 °C.

### 2.5. 2D-DIGE Image Analysis

For protein spot detection, 2D-DIGE gels were scanned with three different wavelengths of the particular CyDye at a resolution of 100 µm using a Typhoon 9410 Scanner (GE Healthcare, Uppsala, Sweden). Gel images were analysed via the DeCyder™ software (version 7.2, GE Healthcare, Uppsala, Sweden). Spots were matched to a master 2D-DIGE gel (a representative pH 4–7 platelet protein map of the IS images). On average, 500 protein spots were matched manually to the master gel using the DeCyder™ software. Afterwards an automatic spot match was used which achieved an average of 4310 matched spots per gel. The standardised abundance (SA) of every protein spot was calculated by the DeCyder™ software with two normalisation steps. The first step is the in-gel normalisation by dividing each spot with the centre volume of the corresponding spot map, and the second one is dividing each normalised spot volume against the corresponding spot normalised spot value of the IS [24]. Detailed information about the image analysis was published by Winkler et al. [25].

### 2.6. Protein Identification via Mass Spectrometry

For MS-based identifications, 250 µg unlabelled proteins were separated by 2D-DIGE and proteins were visualized by MS-compatible silver staining [26]. Protein spots of interest were excised manually from the gels, de-stained, disulphide was reduced as well as derivatized with iodoacetamide and the proteins were tryptically digested. Subsequently, these peptide samples were applied onto a Dionex Ultimate 3000 RSLC nano-HPLC system (Thermo Scientific) and afterward directly subjected to a QqTOF mass spectrometer oTOF compact (Bruker Daltonics) equipped with a nano-flow CaptiveSpray ionisation device. Detailed analytical conditions were previously described [27]. The protein identification was obtained with database searches against UniProtKB/Swiss-Prot (2020-06) using Mascot v2.7 server.

### 2.7. One- and Two-Dimensional Western Blot Analysis

For one-dimensional Western blot (1-D WB), 12 µg total platelet proteins were mixed with a sample buffer (150 mM Tris-HCl pH 8.68, 7.5% SDS, 37.5% glycerol, bromine phenol blue, 125 mM DTT) to obtain a final volume of 20 µL. Afterward, samples were boiled for 4 min at 95 °C and centrifuged for 3 min at 20,000× *g*. Thereafter, the samples were separated on a 11.5% SDS gel (50 V, 20 min and 100 V, 150 min) and blotted (75 V, 120 min) on a nitrocellulose membrane (NC; Pall, East Hills, NY, USA) or polyvinylidene difluoride membrane (PVDF; FluoroTrans^®^ W, Pall, East Hills, NY, USA). For protein quantification, total protein on the membrane was stained using ruthenium-(II)-tris-(bathophenanthroline disulphonate) (RuBPS; dilution 1:100,000 overnight at 4 °C; Sigma-Aldrich St. Louis, MI, USA).

For two-dimensional Western blot (2-D WB) analysis, 36 µg of Cy2-labelled platelet proteins were separated by IEF on either a 7 cm pH 3–10 or a 24 cm pH 4–7 IPG strip as described for 2D-DIGE gels, and subsequently transferred onto an NC or PVDF membrane (75 V, 90 min). Afterward, the membrane was blocked with 5% non-fat dry milk (Bio-Rad, Hercules, CA, USA) in 1x PBS containing 0.3% Tween-20 (PBS-T) overnight at 4 °C under gentle shaking. On the next day, membranes were washed (3× with PBS-T for 5 min, each). For detection, the following primary antibodies were used in the corresponding dilutions by incubation for 2 h at room temperature (180 rpm) in PBS-T containing 3% non-fat dry milk: monoclonal anti-factor XIIIa (F13A1) (ab1834; Abcam, Cambridge, UK, USA) 1:250, polyclonal anti-CD41/integrin alpha 2b (ITGA2B) (ab83961; Abcam, Cambridge, UK) 1:200, monoclonal anti-integrin beta 3 (ITGB3) clone EPR2342 (ab119992, Abcam, Cambridge, UK) 1:1000, monoclonal anti-calreticulin (CALR) clone FMC 75 (ab22683; Abcam, Cambridge, UK) 1:500, polyclonal anti-endoplasmic reticulum chaperone BiP (HSPA5) (ab21685; Abcam Cambridge, UK) 1:250 and monoclonal anti-talin (TLN1) clone 8D4 (SAB4200694, Merck, Germany) 1:300. After washing (3× with PBS-T for 5 min, each), the membranes were incubated with a DyLight 650-conjugated secondary antibody (Novus Biologicals, Littleton, CO, USA), diluted 1:500 or with a horse-radish peroxidase (HRP)-conjugated secondary antibody, diluted 1:20,000 in PBS-T containing 3% non-fat dry milk for 1.5 h in the dark at room temperature (65 rpm). After washing (3× with PBS-T for 5 min, each), the membranes were incubated for 1.5 h in the dark at room temperature (65 rpm). After washing (2× with PBS-T, 1× with 1× PBS for 5 min, each), the antibody fluorescence signals were detected with a Typhoon FLA 9500 imager (GE Healthcare, Uppsala, Sweden) at a resolution of 100 µm. The HRP signal was detected using an Enhanced Chemiluminescent (FluorChem^®^ HD2, Alpha Innotech, San Riandro, CA USA). The 1-D WB antibody signals of F13A1 were normalized by the RuBPS fluorescence signal from the 40 to 100 kDa bands and quantified with ImageQuant 8.0 (GE Healthcare, Uppsala, Sweden).

### 2.8. Measurement of Haemostatic Biomarkers in Plasma

FXIII activity in citrate plasma was measured by Berichrom Factor XIII chromogenic determination (Siemens Healthcare, Erlangen, Germany) according to manufacturers’ instructions. Factor VIII activity was measured on a Sysmex CA 7000 analyzer using factor VIII-deficient plasma (Technoclone) and APTT Actin-FS (Dade Behring). D-dimer levels were measured by a quantitative latex assay (STA-LIAtest D-DI; Diagnostica-Stago, Asnieres, France) on a STAR analyser (Diagnostica-Stago) according to the manufacturer’s instructions. Fibrinogen was routinely measured in platelet-poor plasma according to Clauss (STA Fibrinogen; Diagnostica-Stago, Asnieres, France; normal range: 180–390 mg/dL). C-reactive protein (CRP) was measured with fully automated particle-enhanced immunonephelometry (N high-sensitivity CRP, Dade Behring, Marburg, Germany). For sCD62P, to obtain definitely platelet-free plasma, a second centrifugation step (Eppendorf, Hamburg, Germany) at 13,400× *g* for 2 min was performed. These plasma aliquots were also stored at −80°C, until the determination of sCD62P plasma levels in series. Soluble CD62P levels were measured using a human sCD62P immunoassay (R&D Systems, Minneapolis, MN, USA) following the manufacturer’s instructions.

### 2.9. Fluorogenic F13A1 Activity Assay from Platelet Samples

Washed platelets from 42 patients’ samples were lysed by 1% Triton X-100 (Sigma-Aldrich, St. Louis, MO, USA) and sonicated for 1 min at 4 °C. After centrifugation (15,000× *g*, 4 °C) lysed platelets were transferred to a fresh tube. Protein concentration in the lysate was determined by Coomassie. F13A1 activity in platelets was performed by a fluorogenic FXIIIA enzyme activity kit (Zedira GmbH, #F001, Darmstadt, Germany) essentially according to the manufacturers’ protocol with minor modifications. Measurements were performed in triplicate in 96-well flat-bottom black microplates (Thermo Scientific, #137101, Denmark). Ten µL of lysed samples was mixed with 90 µL reagent mix solution (modified peptide, 100 NIH Units thrombin, Tris buffer pH 7.5 containing CaCl_2_, NaCl, polyethylene glycol (PEG), glycine methyl ester, clot inhibitor peptide, Heparin antagonist (hexadimethrine bromide) and sodium azide), creating a final volume of 100 µL/well. As a control for specificity of enzymatic F13A1 activity, the irreversible transglutaminase inhibitor T101 (50 µM final concentration) was used. The enzyme kinetics were recorded at 37 °C using a Varioskan LUX microplate reader (Thermo Fisher Scientific) with excitation at 313 nm and emission at 418 nm in the kinetic mode, absorbance was read every 36 s for 15 min and the change of absorbance between 0 and 15 min was detected. F13A1 activity was determined on the basis of a standard curve constructed with washed human platelets. Microsoft Excel was used for further processing of enzyme kinetic data.

### 2.10. Plasmin–α-2-Antiplasmin (PAP) Complex Quantification in Plasma by ELISA

Quantification of PAP complexes in plasma samples was performed using the Technozym PAP Complex ELISA Kit (Technoclone, Vienna, Austria) according to the manufacturer´s instructions.

### 2.11. Protein Disulfide Isomerase (P4HB) Quantification in Plasma by ELISA

Quantification of P4HB in plasma samples was performed using the Human PDI/P4HB ELISA Kit (LSBio LifeSpan BioSciences, Inc., Seattle, WA, USA) according to the manufacturer´s instructions.

### 2.12. Biological Pathway Analysis

Biological data base analysis was made from the fifteen lung cancer-related platelet proteins. The data source for the protein–protein interaction (PPI) networks was the protein query of the STRING database [28], with the settings: active interaction sources, experiments and databases; score = 0.4; maximal additional interactors = 0. For the functional enrichment, the Gene Ontology Biological Processes and KEGG pathway analysis were used with a specific colour for each biological process and KEGG pathway. The STRING Version 11.0b was used.

In addition, we applied the NetworkAnalyst platform [29,30] for further analysis. The differentially expressed proteins were uploaded to the web-platform (www.networkanalyst.ca, accessed on 13 January 2021) and generic protein–protein interaction (PPI) network analysis was performed with the Imex interactome [31] resulting in a first-order network with 722 nodes and 889 edges comprising the 15 lung cancer proteins as seeds. This network was downloaded as a graphML file and imported into the Cytoscape 3.8.2 program. The stringApp was used to “STRINGify” the network, followed by functional enrichment of pathways and gene ontologies, which were sorted according to *p*-values. Relevant pathways were selected and coloured using the bypass function for node colours.

### 2.13. Statistics

For statistical analysis, from each 2D-DIGE image only protein spots were included which could be matched by the IS spot map with more than 95% of all 2-D platelet proteome maps of this study. This quality selection limited resulting protein spots to 566 from an average 2720 to the master gel-matched spots. One-way and two-way analysis of variance (ANOVA) were calculated for these 566 highly reliable matched spots between the three study groups (healthy controls, patients with lung and brain cancer). Resulting *p*-values of the one-way ANOVA were false discovery rate-corrected (FDR) by Benjamini–Hochberg for multiple comparisons [32]. FDR-corrected one-way ANOVA and unadjusted two-way ANOVA were calculated with the Extended Data Analysis (EDA) module of the DeCyder™ software (version 7.2, GE Healthcare, Uppsala, Sweden). Supervised principal component analysis (PCA) of one-way ANOVA significant spots was also performed by this EDA module to control sample clustering. To assess the effect of mortality on the examined platelet proteome between cancer type groups, the two-way ANOVA main effects “mortality” and “cancer type group” and the interaction term “cancer type group ∗ mortality” were included. Significant differences between the control group and each cancer group were analysed by planned post-hoc contrasts analysis with unpaired Student´s t-test in SPSS and corrected for multiple comparison by the online FDR calculator [33]. For a direct comparison of different parameters the effect size was calculated by Cohen’s D = (mean^1^ − mean^2^)/standard deviation^pooled^). Graphs were created with GraphPad Prism 6 (GraphPad Software, Inc. San Diego, CA, USA).

### 2.14. Control Evaluation of the Platelet Proteome Analysis in Comparison with the Different Platelet Counts between Study Groups

Platelet counts of patients with lung cancer were significantly higher compared to healthy controls and patients with brain cancer (Table 1). Hence, it may be supposed that changes in the platelet proteomes of patients with lung cancer can also be associated with increased platelet counts. To prove this possible bias of a platelet proteomics study, platelet counts were correlated with the 566 2D-DIGE quantified platelet protein spots from the whole study cohorts. No significant correlation could be detected by these statistical control evaluations. This assessment ensured that cancer-related protein profiles in platelets could not be biased by differences in platelet counts.

## 3. Results

### 3.1. Patient Characteristics

To investigate if and how the platelet proteome is affected in patients with cancer and high thrombosis risk, in total 82 study participants were included in this study: twenty-two patients with brain cancer, 19 patients with lung cancer and 41 healthy age- and sex-matched controls.

Both brain and lung cancer patients had significantly higher plasma FVIII, fibrinogen and D-dimer levels compared to healthy sex- and age-matched controls. Antithrombin III activity was significantly increased in patients with brain cancer.

During the course of the study, two patients with brain (9.1%) and two patients with lung cancer (10.5%) had a VTE. These thrombotic events included pulmonary embolism (PE; 7.3%) and deep vein thrombosis (DVT; 2.4%). Eight patients with brain cancer (36.4%) and eleven patients with lung cancer (57.9%) died during the follow up (Table 1). These detailed characteristics of the study populations are presented in Table 1.

### 3.2. Platelet Proteome of Patients with Brain and Lung Cancer Compared to Controls

Figure 1 shows a schematic representation of the applied workflow.

For the identification of differentially regulated platelet proteins in patients with brain and lung cancer and matched healthy controls, the platelet proteome of these samples were analysed by 2D-DIGE in the pH range 4–7 (Figure 2). After the application of protein spot quality selection criteria (defined in the Material and Methods section) a total of 566 protein spots were included in statistical analysis.

One-way ANOVA of the platelet proteome revealed 36 significant protein spot changes between the three study groups (FDR-adjusted *p* < 0.05; Figure 2; Table 2). Identification of these 36 protein spots by MS showed that these proteoforms corresponded to 19 different proteins (Table 2). Adjusted post-hoc contrast calculations revealed 27 significant lung cancer-related protein spot changes but no significant brain cancer-related difference (Table 2). Identification of the protein spots by MS showed that the 27 lung cancer-related proteoforms belonged to 15 different proteins (Table 2). This differential platelet proteome of the cancer groups and the healthy controls was also clearly visible graphically by a PCA analysis (Appendix A). The entire raw data of this platelet proteome analysis can be found in the Appendix A.

### 3.3. Pathway Analysis of Lung Cancer-Related Platelet Proteins

To assess the coherent protein networks and functional relationships of lung cancer-related platelet protein changes, biological database analysis was conducted. Unbiased STRING protein interaction analysis allocated lung cancer-related changes mainly to proteins primarily responsible for thrombosis. Based on the altered levels of ITGA2B, ITGB3, TLN1, F13A1, TF and HSPA5, the highest significant functional enrichment was found for the biological process of platelet degranulation (*p* = 3.19^−11^) and for platelet activation in KEGG pathways (*p* = 4.47^−5^). Additionally, a significantly altered KEGG pathway was protein processing in the endoplasmic reticulum (ER) (*p* = 4.74^−6^) (Figure 3). These findings were further supported by analyses using the *NetworkAnalyst* approach [29] (Appendix A).

Based on these findings and literature search the main stabilizer of the fibrin clot, F13A1, and the ER chaperones, P4HB, CALR and HSPA5, important for folding and glycosylation of secretory proteins such as coagulation factors [34,35], were selected for in-depth characterisations and functional explorations.

### 3.4. F13A1 Protein Processing Is Changed in Platelets of Patients with Lung Cancer

The abundance of a 55 kDa fragment (pI 5.00) from F13A1 was significantly increased in platelets of patients with lung cancer (Table 2; Figure 2). The fibrin crosslinking protein F13A1 is a transglutaminase with a molecular weight of 83 kDa.

Generally, F13A1 is activated by thrombin with the enzymatic removal of a short (2–37) activation peptide [36,37], which results into the 79 kDa activated F13A1 proteoform, whereas a 55 kDa fragment of F13A1 is observed to be produced after the enzymatic inactivation of activated F13A1 by further cleavage with, e.g., thrombin [38], chymase [39] and plasmin [40]. MS analysis of the lung cancer-related 55 kDa F13A1 cleavage product detected peptides between the amino acid sequence 13 and 492 (Appendix A). For inactivation of F13A1, thrombin cuts at K514 [38], chymase at F574 [39] and plasmin at R492 [40]. Accordingly, several of these serine proteases can be responsible for this 55 kDa F13A1 cleavage product in platelets (Appendix A). Interestingly, MS analysis of this platelet 55 kDa F13A1 fragment also identified peptides from the amino acid sequence 13 to 38, which is within the amino sequence of the activation peptide. Thus, the present MS data indicate that this 55 kDa fragment was not cleaved from the activated proteoform of F13A1 in platelets (Appendix A).

A long-standing hypothesis is that activated F13A1 is inactivated by the fibrinolytic system [40]. To evaluate these in vitro observations in the clinical situation of this study, the levels of the 55 kDa fragment of F13A1 in platelets of patients with lung cancer and matched controls were correlated with the corresponding plasma levels of the fibrinolytic markers plasmin–α-2-antiplasmin complex and D-dimer. The plasmin–α-2-antiplasmin complex represents the peripheral marker of in vivo plasmin generation. D-dimer is the degradation product of F13A1-crosslinked fibrin and is therefore a routinely established indicator for VTE and also reflects the risk of cancer-associated thrombosis [41]. The plasmin–α-2-antiplasmin complex was significantly increased (FC = 2.4; **p_unadj_** = 4.60^−5^; Appendix A) in the plasma of patients with lung cancer, and the significant correlation with the 55 kDa fragment of F13A1 in platelets (r_s_ = 0.451; *p_unadj_* = 4.44^−3^; Figure 4a) supports the theory of plasmin-mediated degradation of F13A1. In addition, the significant correlation of this 55 kDa fragment of F13A1 with the D-dimer (r_s_ = 0.599; *p_unadj_* = 1.40^−4^; Figure 4b) indicates the interplay between F13A1-mediated crosslinking of fibrin and plasmin-mediated fibrinolysis in patients with lung cancer.

Two-dimensional Western blot analysis was carried out for the identification of further proteoforms from F13A1 in the platelet proteome. An antibody, raised against the full-length F13A1, recognised the respective 55 kDa proteoform. In addition, the F13A1 antibody detected three spots with a MW of 83 kDa and pI between 5.85 and 5.65. Subsequent MS analysis confirmed these spots as F13A1 proteoforms. Another two F13A1 spots (12 and 12d) were identified by MS in the near neighbourhood (Appendix A). The abundance of all these full-length F13A1 spots was also found to be increased in platelets of patients with lung cancer compared to matched healthy controls, except one spot with a pI of 5.85 (spot 12c; Table 3). Increased levels of the 55 kDa F13A1 proteoform in platelets of patients with lung cancer were confirmed by fluorescence 1-D WB with a significant correlation (r_s_ = 0.843; *p* = 2.0^−6^) with the corresponding 2D-DIGE signals (Appendix A). The abundance of the full-length F13A1 proteoforms were visible as a single band but the lung cancer-related increase in the 83 kDa proteoforms between the pI 5.75 to 5.60 could not be detected by 1-D WB. These unchanged quantitative 1-D profiles of the full-length F13A1 in patients with lung cancer may be based on the significant negative (opposite) correlation of the abundance from the most alkaline F13A1 spot (pI 5.85; spot 12c) with the levels of all other 83 kDa proteoforms from F13A1 (Appendix A). Interestingly, this F13A1 spot 12c with the pI 5.85 was also changed in the opposite direction in patients with lung cancer compared to all other F13A1 proteoforms (Table 3).

To investigate the functional outcomes of these altered levels of F13A1 proteoforms in patients with lung cancer, enzymatic activity was measured in the platelets and plasma of the particular study samples. No change in enzymatic F13A1 activity was detectable in platelets of patients with lung cancer (Figure 4c). However, enzymatic F13A1 activity significantly correlated with the levels of F13A1 spots 12b (pI 5.75), 12 (pI 5.60) and 12a (pI 5.65), the abundances of which were significantly increased in patients with lung cancer (Table 3). Once again, the F13A1 level of spot 12c (pI 5.85) attracted attention in the opposite direction by a weak trend of a negative correlation with enzymatic F13A1 activity (Table 3). These observations indicate that a pI-shifting PTM regulates the enzymatic activity of these 83 kDa F13A1 proteoforms in platelets, whereby the most alkaline one (spot 12c; pI 5.85) seems to be the inactive variant. The 55 kDa spot 11 of F13A1 did not correlate with the enzymatic activity of this transglutaminase (Table 3).

Because in clinical studies F13A1 is more frequently investigated in plasma, its enzymatic activity was also analysed in the plasma of recruited cancer patients and matched healthy controls. Plasma F13A1 activity was unchanged in patients with lung cancer and was significantly decreased in patients with brain cancer (Figure 4d; Appendix A).

### 3.5. Chaperones from the KEGG Pathway “Protein Processing in ER” Are Elevated in Platelets of Patients with Lung Cancer

One-way ANOVA-based differential 2D-DIGE analysis identified significantly changed abundances of two P4HB, five CALR spots and one spot of HSPA5 in platelets between patients with lung and brain cancer and matched healthy controls (Table 2). Post-hoc contrast analysis showed that all these ER chaperones were increased in platelets of patients with lung cancer. Two-dimensional WB analysis confirmed MS identifications of these eight CALR and two HSPA5 (Appendix A). The MS identification of P4HB spots in platelets we already validated in a previous platelet proteomics study of LA patients [16]. In this study we also detected elevated P4HB levels in the plasma of LA patients with thrombosis history. In the actual study, P4HB concentration was not significantly increased in the plasma of patients with lung cancer (Appendix A).

### 3.6. Functional Relationships of the Platelet Proteome with Haemostatic Plasma Laboratory Parameters from Patients with Lung Cancer and Matched Controls

Interestingly, ER chaperones are frequently described to assist in the final biosynthesis of coagulation factors. The main member P4HB [42] of the protein disulfide isomerase (PDI) family mediates the formation of disulphide bond from FIII (tissue factor) [43,44,45], FV [46] and FXI [47]. This redox-based change of their conformation contributes to enhanced activation of these coagulation factors and consequently to an accelerated generation of thrombin [48] and fibrin. Although plasma D-dimer is an indicator for enhanced fibrinolysis, increased D-dimer levels also indicate the preceding step of an increased fibrin generation. Thus, the functional enhancement of coagulation factor activity (e.g., FV) by more PH4B, may be indirectly reflected by the significant correlation of platelet PH4B levels with the plasma D-dimer levels of patients with lung cancer and matched controls (spot 28: r_S_ = 0.586; *p_unadj_* = 3.10^−6^).

The ER proteins CALR and HSPA5 are essential chaperones for the glycosylation and secretion of proteins and this function is described for the final biosynthesis of coagulation factors [35,49,50]. Besides F13A1, platelets are also a potential source for coagulation factors, e.g., FV [46] and FVIII [51,52,53]. Hence, the FVIII activity, an important activation marker of the haemostatic system, was significantly increased in the plasma of patients with lung cancer (Table 1; FC = 1.75; *p* = 0.0002) and to a lesser extent in patients with brain cancer (Table 1; FC = 1.26; *p* = 0.001). In line with the assumption that the processing and release of this glycoprotein may be influenced by the enhanced pathway of “protein processing in ER”, a significant positive correlation was detected between the ER-chaperoning proteins’ CALR spot abundances (spot 3: r_S_ = 0.557; *p_unadj_* = 3.41^−4^) and HSPA5 (spot 15: r_S_ = 0.628; *p_unadj_* = 3.23^−5^) of platelets with the corresponding plasma values of FVIII from patients with lung cancer and matched controls (Figure 5a,b).

However, these observations of significant correlations of platelet ER glycosylation chaperones with FVIII or the 55 kDa F13A1 fragment and P4HB with plasma D-dimer levels may be biased by literature-based hypothesis generation. To uncover and manifest an unbiased and specific interaction of certain platelet proteins with haemostatic and inflammatory plasma biomarkers, all 566 2D-DIGE-quantified platelet proteoforms from patients with lung cancer and matched controls were correlated with the corresponding plasma levels of five well-known cancer-associated VTE plasma biomarkers. These were FVIII activity [54,55], prothrombin fragment 1 + 2 [56], D-dimer [3], fibrinogen [57] and the acute phase protein CRP. In fact, the strongest correlations of plasma FVIII with these 566 platelet proteins were found with the ER-chaperoning proteins P4HB, HSPA5 and CALR allocated to the interactome pathway “protein processing in ER” (Figure 6d). Remarkably, the substrate for F13A1, plasma fibrinogen, had the strongest correlation with the 55 kDa fragment of fibrin stabilising factor F13A1 of the platelet proteome (Figure 6e). The platelet 55 kDa fragment of F13A1 also had one of the strongest correlations with the acute phase proteins CRP (Figure 6c), which indicates the strong interplay of inflammation with the essential steps of haemostasis, which finally may result into immunothrombosis. In contrast with the strong association of plasma FVIII with the pathway “protein processing in ER” in platelets, D-dimer, fibrinogen and CRP correlated significantly with proteins from platelet degranulation and platelet activation pathways such as filamin A (FLNA) and the integrins ITGA2B and ITGB3 (Figure 6a,c,e). No strong correlations of the platelet proteome were found with the plasma laboratory values of prothrombin fragment 1 + 2, antithrombin III and plasminogen activator inhibitor 1.

For additional functional proof, for the specificity of lung cancer-related platelet protein profiles to platelet activation, the 566 2D-DIGE-quantified platelet proteoforms were also correlated with plasma levels of sCD62P, a reliable biomarker for platelet activation in vivo [58]. Again, platelet degranulation and the ER protein processing pathways were significantly enriched via the sCD62P-correlating platelet proteins by STRING interactome analysis (Figure 6b). All platelet proteins that correlated significantly with haemostatic plasma laboratory parameters are indicated in Appendix A.

The MS identifications of ITGA2B, ITGB3 and TLN1 were also verified by 2-D WB analysis. These immunological platelet proteome analyses also revealed several additional proteoforms of the respective proteins (Appendix A).

### 3.7. Associations in the Platelet Proteome with Lung Cancer and Risk of Death

Further explorative endpoints of this platelet proteomics study were occurrence of VTE or death during two year follow up, respectively. Of 41 patients with lung or brain cancer, eighteen died and four developed VTE. To study associations of the platelet proteome with patient outcome the endpoint mortality was selected. An explorative two-way ANOVA analysis and planned post-hoc contrast analysis of the 566 investigated platelet proteins revealed that SERPINB1 (leukocyte elastase inhibitor) had the strongest dependency (*p* = 0.028) on lung cancer and mortality. Previously, we identified that in LA patients a decreased expression of SERPINB1 favoured prothrombotic neutrophil extracellular traps formation [16]. The quantitative representation of the actual group effects separately for cancer type and mortality showed that SERPINB1 was significantly decreased in deceased compared to surviving patients with lung cancer (FC = 0.82; *p* = 0.03) and healthy controls (FC = 0.83; *p* = 0.003). SERPINB1 abundance was independent of mortality in patients with brain cancer (Figure 7).

For a small explorative follow up case-report, we performed 2D-DIGE analysis of platelet samples from four patients with lung cancer after three and six months to evaluate how the identified changes in protein profiles in lung cancer are associated with patient outcome. These patients were given chemotherapy within 3 months of the first blood draw. One of the traced patients had a PE at two months after the baseline and died 5.5 months after study inclusion.

The platelet SERPINB1 level of this deceased patient stayed decreased (SA = 1.10) after three months compared to the corresponding baseline value (SA = 1.17) and the mean baseline value of all deceased patients (SA_mean_ = 1.10; *n* = 11) with lung cancer (Appendix A). After three months, the SERPINB1 levels decreased also in the surviving patients but returned almost to baseline values after six months and thus almost aligned with the mean values of healthy controls.

Quite similar patterns in the follow up tracing were observed for the lung cancer-related proteins ITGA2B, PH4B and 55 kDa F13A1. Thus, these lung cancer-related alterations were most pronounced in the deceased patient, who suffered from PE two weeks before the second blood sampling (3 months value). In the surviving patients, the lung cancer-related changes were also fortified after 3 months and returned almost to baseline values after 6 months (Appendix A). However, exclusively, the abundance of SERPINB1 was already significantly associated with patient outcome at baseline.

## 4. Discussion

Continuous platelet activation may contribute to increased VTE risk in patients with brain and lung cancer. To the best of our knowledge, this is the first study comparing the platelet proteome of these patients and matched healthy controls to elucidate prothrombotic pathways. These investigations showed a predominant influence of lung cancer on the peripheral platelet proteome compared to brain tumours. Altered platelet protein levels in patients with lung cancer could be associated with increased platelet degranulation, changes in the conversion of F13A1 and upregulated protein processing in ER; the latter is believed to be functionally linked to increased FVIII plasma levels. The identification of these protein changes in platelets expands our knowledge of differential molecular mechanisms for the prothrombotic state in lung and brain cancer.

Interestingly, no brain cancer-related changes were detected by the 2D-DIGE analysis in the platelet proteome. However, in a previous study we identified that the high podoplanin expression of brain tumours is functionally linked with the increased VTE risks of this cancer type [59]. For the functional examination of this assumption, it is shown that podoplanin on primary human glioblastoma cells induces platelet activation via the specific binding to their C-type lectin receptor type 2 receptors. Since increased intra-tumoral platelet aggregates correlate with VTE events in patients with brain cancer [59], it can be reasoned that platelets are caught by podoplanin of the brain tumour and related proteome changes are not detectable in peripheral blood platelets.

For pathophysiological explorations of platelet proteome changes in patients with lung cancer we concentrated on four candidates described to be involved in the processing of coagulation proteins, which are F13A1 and the ER proteins P4HB, CALR and HSPA5. An altered conversion of the final clotting factor F13A1, allocated to the pathway “platelet degranulation”, was detected by elevated levels of a 55 kDa cleavage product from F13A1 in the platelets of patients with lung cancer. The fibrin-stabilizing factor F13A1 is well known to circulate in the plasma as a heterotetrameric protein complex, consisting of a dimer of the catalytic F13A1 subunits and a dimer of the carrier/inhibitory FXIIIB2 subunits. The transglutaminase F13A1 is a critical determinant of venous thrombus characteristics such as thrombus size, stability and red blood cell retention [60,61]. Platelets contain only F13A1. Although the concentration of F13A1 is around 150-fold higher in platelets compared to plasma [62], its functions in platelets are quite unknown. Only recently it was shown that platelet surface-exposed F13A1 contributes to thrombus stability, whereas the secreted F13A1 from platelets is functionally irrelevant [63]. However, the enzymatic activity of F13A1, exposed on platelets, is temporally limited, with a maximum after 5 minutes, and completely declines after one hour [63]. The mechanisms of F13A1 inactivation are unclear and of substantial interest [64]. Our findings that the platelet level of the 55 kDa fragment of F13A1 did not correlate with the enzymatic activity of this transglutaminase in platelets suggests that it may be a product of accelerated F13A1 inactivation in patients with lung cancer. The correlation of the platelet 55 kDa fragment of F13A1 with the plasma plasmin–α-2-antiplasmin complex and D-dimer would suggest an inactivation by plasmin and thus the involvement of the fibrinolytic system. So far, however, it has only been reported that this protease exclusively inactivates thrombin-activated F13A1. Whereas, the actual platelet proteomics study showed that the 55 kDa lung cancer-related F13A1 fragment originated from its precursor, since peptides from the activation region were also detected by MS analysis. The finding of an inactivation of the F13A1 precursor in platelets becomes functionally reliable in the context of previous investigations, showing that the enzymatic activity of F13A1 can also be initiated in platelets by the increase in intracellular Ca^2+^ without the need of thrombin-mediated activation [65].

The regulations of structural changes of F13A1 during enzymatic activation are less characterized. The most striking phenomenon during the conformational change of activated F13A1 is a high number of lysine sides becoming accessible for acetylation in the core domain of the enzyme [66]. The current 2D-gel-based proteomics study showed that all acidic 83 kDa proteoforms of F13A1 had a significant positive correlation with the enzymatic activity of this transglutaminase except the most highly abundant alkaline one. These unique observations including the negative association with enzymatic activity of this latter F13A1 proteoform, fortifies the hypothesis that acetylation may cause these shifts in their pIs and the regulation of enzymatic F13A1 activity in platelets.

However, no significant change in F13A1 enzymatic activity could be measured in platelets and plasma of patients with lung cancer. Although F13A1 abundance was unchanged in platelets of patients with brain cancer, a significant decrease in enzymatic F13A1 activity was found in the plasma of patients with brain cancer. Decreased plasma levels of F13A1 were previously observed in the plasma of patients with thrombosis history with or without cancer. It is assumed that this decrease is caused by an increased consumption of F13A1 due to its inactivation [67]. In the actual study, the increased levels of the precursor and the 55 kDa fragment from F13A1 may also indicate an increased turnover of F13A1 in the platelets of patients with lung cancer.

Platelets are also of functional importance for the progression of cancer and metastasis formation [68]. The observed increased levels of F13A1 and changed processing of F13A1 in platelets of patients with lung cancer may also play a role in metastasis. In F13A1-deficient mice, the metastatic potential was significantly diminished. F13A1 was shown to support metastasis primarily by limiting natural killer cell-mediated clearance of micrometastatic tumour cells [69]. F13A1 has been shown to be associated with inferior outcomes in acute promyelocytic leukemia [70] and the activity of F13A1 in plasma was elevated in patients with non-small cell lung carcinoma [71]. High expression of F13A1 was also observed in inflammatory monocytes of patients with lung squamous carcinomas, which promote fibrin cross-linking and in turn facilitate metastases of lung carcinoma cells [72].

Elevated levels of ER chaperones P4HB, CALR and HSPA5 indicated the induction of the ER unfolded protein response (UPR) in platelets of patients with lung cancer. An increased abundance of these ER proteins was already previously observed in platelets of LA patients with thrombosis history [16].

Interestingly, the UPR in ER is also associated with malignant transformation of cancer cells [73] as well as with a prothrombotic influence of pancreatic cancer. In this particular study, it was shown that proteins of the UPR in cancer are released from pancreatic cancer cells by extracellular vesicles. Increased levels of the same UPR protein profiles are also detected in the plasma of patients with pancreatic cancer and thrombosis history. These observations indicate a mechanistic link between tumour progression and cancer-associated thrombosis [74]. That group also showed that inhibiting enzymatic PDI activity in plasma and platelets by the flavonoid isoquercetin reduced the hypercoagulability of advanced cancer patients [75], as well as that of LA patients [46]. Remarkably, in patients with lung cancer as well as in LA-positive patients with thrombosis history [16] we identified a highly significant increase in the PDI family member, P4HB, in platelets, which also manifest this ER chaperone as a highly purposive antithrombotic drug target. In the current study, however, the plasma concentrations of P4HB were not increased in patients with lung cancer.

The ER proteins HSPA5 and CALR mediate to a great part the glycosylation and folding of secretory proteins. Interestingly, the function of these ER proteins and the induction of the UPR in the ER have been extensively investigated and described for the industrial and therapeutic production of FVIII [49,76,77]. Increased levels as well as increased activity of FVIII in plasma are generally associated with high VTE risk [78] and predict VTE in cancer patients [54]. In the current study, FVIII was also significantly increased in the plasma of patients with lung cancer and showed a predominant correlation with the ER proteins HSPA5 and CALR in comparison to all other proteomic investigated platelet proteins. This induction of the UPR and correlations of FVIII with P4HB, HSPA5 and CALR levels may also take place systemically in patients with lung cancer. Thus, platelets may be a peripheral indicator for enhanced glycosylation and secretion from, e.g., liver sinusoidal endothelial cells [79]. The lung cancer-related upregulation of ER chaperones in platelets corroborate a functional association of the UPR with increased FVIII production and thrombosis risks.

The abundance of both integrins of the fibrinogen receptor integrin αIIbβ3, ITGA2B and ITGB3, were significantly reduced in platelets of patients with lung cancer. These glycoproteins are the two integral parts of the key adhesion receptor of platelets and play a crucial role in fibrinogen-mediated thrombus formation by promoting platelet adhesion. Diminished levels of these integrins in platelets do not functionally correspond to an expected increased adhesion and aggregation of platelets in patients with high thrombosis risks. The decrease in ITGA2B and ITGB3 levels in platelets during this prothrombotic condition of patients with lung cancer and LA with thrombosis history [16] may be caused by the continuous shedding of extracellular vesicles, which was recently also described as “membrane pearling” of platelet pseudopodia [80]. In relation to their total protein content, extracellular vesicles of blood platelets have a higher load of ITGA2B and ITGB3 than their corresponding resting platelets [81]. Thus, these integrin levels may be declined in exhausted patient platelets, because of an extensive release of extracellular vesicles. In fact, an increase in platelet-released extracellular vesicles is observed in the plasma of patients with cancer [82] and LA [83].

Accordingly, lower levels of ITGA2B and increased levels of the ER proteins P4HB, CALR and HSPA5 in patients with lung cancer and LA suggest some common affected pathways in their prothrombotic pathology. In contrast to that, the altered proteoforms of F13A1 in platelets appeared to be specific for patients with lung cancer. The data of these two platelet proteomics studies [16] are highly comparable since exactly the same methodology of platelet isolation and 2D-DIGE proteome analysis were used.

Finally, using an exploratory two-way ANOVA that included both cancer type and patient mortality, we found significantly reduced SERPINB1 levels in non-surviving lung cancer patients in the platelet proteome data of the current 2D-DIGE study, but not in brain cancer. SERPINB1 is an antagonist of NET formation. Although SERPINB1 is highly concentrated in platelets, its presence and functionality in platelets to reduce NET formation are heretofore unknown. We have also previously observed that increased NETosis predicts mortality in patients with lung cancer, but not in brain cancer [84]. Subsequently, we found that in LA patients with a history of VTE, decreased SERPINB1 levels in platelets are associated with increased NET formation [16]. Since SERPINB1 is also detected in the secretome of platelets [81], this secretory pathway could explain how the platelet-based SERPINB1 can counteract NETosis in the plasma.

Altogether, our study has both limitations and strengths. The selected proteomics technology of 2D-DIGE, in the pH range 4–7, does not cover the whole possible pH range of 3–10. Low-abundance and high-molecular weight proteins are not captured by 2D-based analysis, meaning a lower sensitivity as compared to MS-based shotgun analyses. Hence, the probability is high that we did not detect all protein changes in platelets of patients with lung and brain cancer. However, there is not a single proteomics technology nowadays that can analyse the whole proteome including all PTMs from a biological sample. PTMs are particularly important for regulating haemostasis precisely and as quickly as possible, e.g., by the PTM "enzymatic cleavage" of coagulation factors. Therefore, 2-D gel electrophoresis should be more versatile to study the prothrombotic phenotype of platelets and their regulation of haemostasis than the "bottom-up" shotgun proteomics currently mainly used. Conventional MS-based shotgun proteomics approaches cannot capture the information of regulatory variation of different proteoforms to each other, due to the necessary pre-analytical proteolytic digestion of biological samples [85]. Accordingly, 2D-DIGE analysis had an analytical advantage in the concrete clinical proteomics study. For example, it would not have been possible to reveal the lung cancer-related changes of F13A1 processing by MS-based shotgun analysis, because the preanalytical digestion of the sample would have eliminated the information of different pIs and MWs of the respective F13A1 proteoforms.

From the clinical perspective, our study is rather explorative, with the inclusion of a relatively small number of patients and matched healthy controls. Due to the limited sample size and number of VTE events it was not possible to perform well-founded statistical subgroup analysis to evaluate comprehensively the influence of VTE or mortality on the investigated platelet proteome.

## 5. Conclusions

Our study delivers novel insights into the differential behaviour of platelets in patients with different types of cancer. These data provide the first evidence that alterations of the platelet proteome are more pronounced in lung cancer compared to brain cancer. Lung cancer-related changes, such as increased platelet levels of P4HB from the PDI family, underscore the importance of this antithrombotic drug target for the hypercoagulable state in certain cancer types. F13A1 might be also involved in the development of VTE, as well as in lung cancer progression, and especially this patient group may benefit in multiple ways from antithrombotic treatments such as FXIII inhibitors. Our clinical platelet proteomics study also confirms previously characterized deregulations of excessive platelet activation in lung cancer and also provides new molecular insights into the thrombosis mechanisms in lung cancer and their correlation with inflammation. These potential antithrombotic drug targets in platelets should be further investigated in cancer research. Some examples of these next steps are that lung cancer-associated platelet proteins should be screened in larger numbers of patients to assess the prognostic potential for thrombotic events and mortality. It is also important to study how the platelet proteome is affected in other cancers with a high risk of thrombosis, such as pancreatic cancer. Furthermore, it should also be compared whether the platelet proteome of patients with acute thrombosis is similar to that of lung cancer, and how it changes over the next 6 months. These observations would also show how directly these platelet biomarkers are functionally linked to thrombosis, and how well they can predict risk.

## Figures and Tables

**Figure 1 cancers-13-02260-f001:**
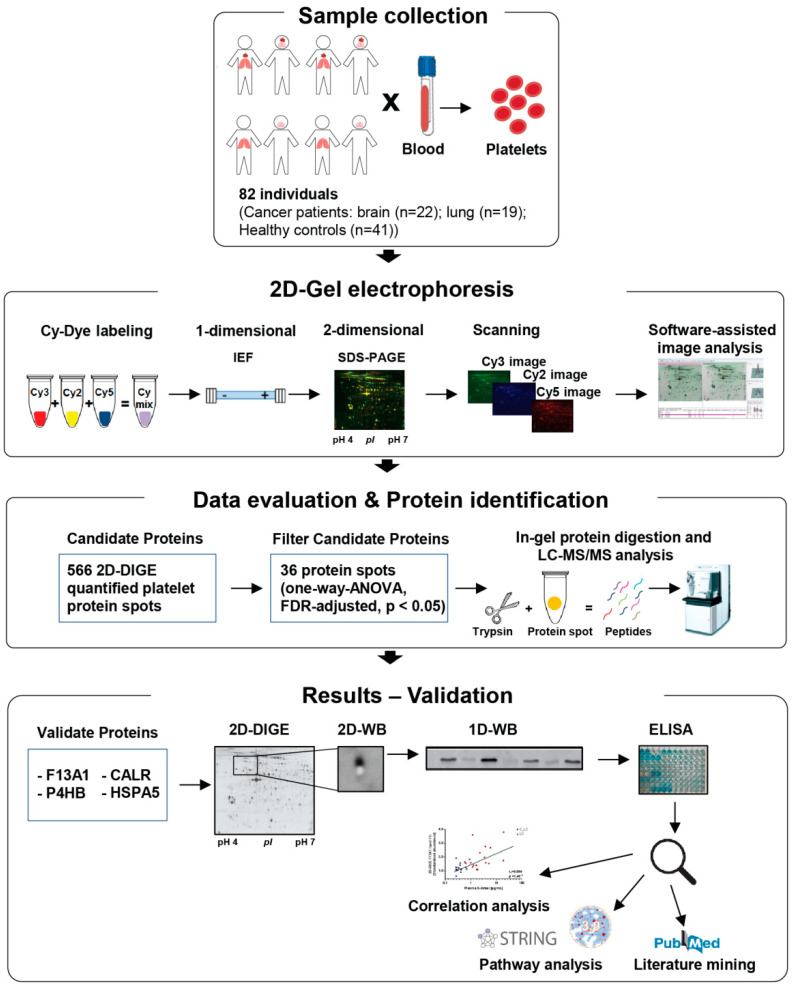
Schematic depiction of the workflow followed in the present study.

**Figure 2 cancers-13-02260-f002:**
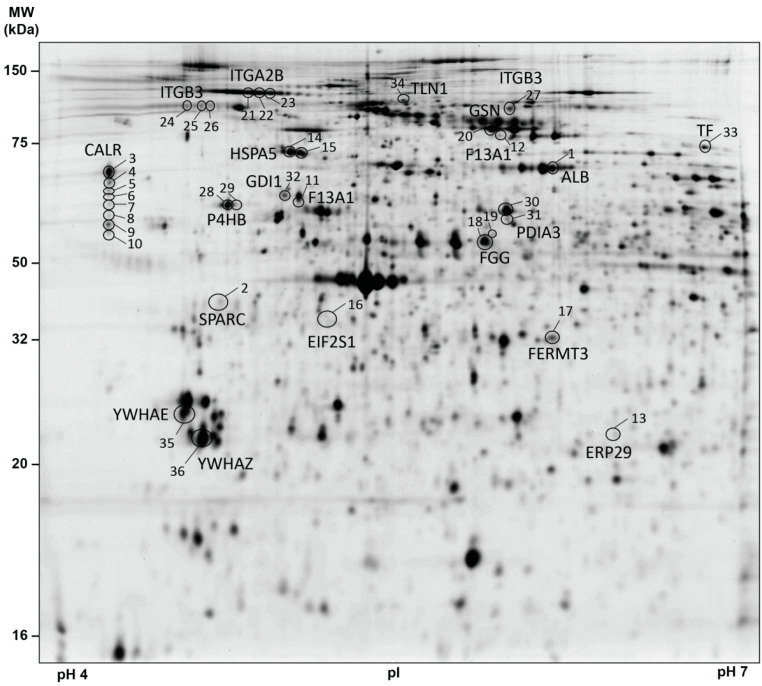
2D-DIGE-based proteome analysis of platelets from patients with brain and lung cancer compared to controls. This representative 2D-DIGE image shows all one-way ANOVA-filtered significantly altered protein spots between patients with lung (*n* = 19) and brain (*n* = 22) cancer compared to healthy controls (*n* = 41). A total of 36 µg (12 µg sample Cy3-, 12 µg sample Cy5- and 12 µg IS Cy2-labelled) of platelet protein extracts was separated according to the isoelectric point (pI) in the pH 4–7 range (separation distance 24 cm) and the molecular weight (MW, separation distance 20 cm). Protein spots identified by MS are circled and labelled with their corresponding gene name and spot numbers are given in Table 1. Protein spots of interest were selected according to (a) protein spots matched > 95% of all 2D-DIGE gels and (b) FDR-corrected one-way ANOVA *p*-value < 0.05. Abbreviations: MW—molecular weight; kDa—kilodalton; pI—isoelectric point; 2D-DIGE—two-dimensional differential in-gel electrophoresis; IS—internal standard; MS—mass spectrometry.

**Figure 3 cancers-13-02260-f003:**
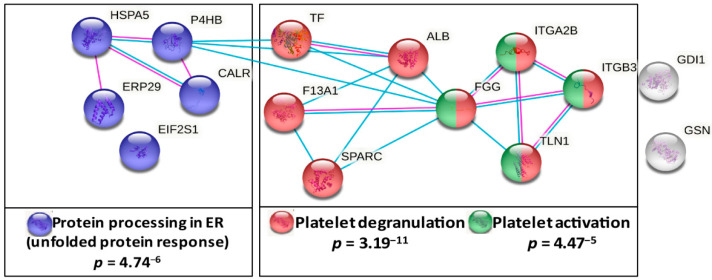
Functional association analysis of lung cancer-related platelet proteins. Network and enrichment analysis shows top pathways obtained upon entering the set of significantly lung cancer-related platelet proteins in the STRING database analysis tool [25]. The type of interaction is indicated by coloured linear slopes; pink: experimentally determined, blue: from curated databases. The enrichment graphs depict the most significantly enriched GO Biological Process with platelet degranulation and the two most significant KEGG pathways in red, in cobalt blue, protein processing in ER and in green, platelet activation. The proteins highlighted in grey were not significantly associated with these quoted functional networks.

**Figure 4 cancers-13-02260-f004:**
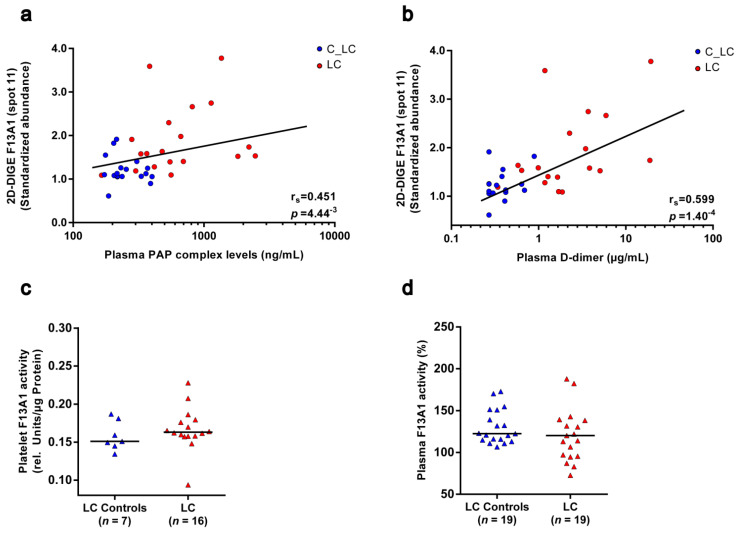
Correlation of the 55 kDa fragment abundance from F13A1 in platelets with the plasma fibrinolysis marker and F13A1 enzymatic activity in platelets and plasma of patients with lung cancer and matched healthy controls. (**a**) Scatter dot plot and correlation analysis of 55 kDa F13A1 standardised platelet protein spot abundances quantified by 2D-DIGE with corresponding plasma PAP complex levels and (**b**) plasma D-dimer levels. An association was assessed by a Spearman´s rank correlation coefficient (r_s_). (**c**) F13A1 activity levels in platelets and (**d**) plasma in patients with lung cancer and matched healthy controls. Protein levels were depicted as single values and median. Abbreviations: 2D-DIGE—two-dimensional differential in-gel electrophoresis; LC—patient with lung cancer; C—healthy control; PAP—plasmin–α-2-antiplasmin.

**Figure 5 cancers-13-02260-f005:**
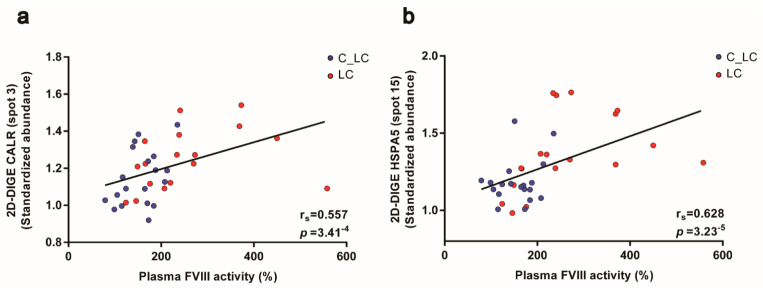
Platelet CALR and HSPA5 abundance and their correlations with plasma FVIII activity from patients with lung cancer and matched healthy controls. (**a**) Scatter blot and correlation analysis of CALR spot 3 and (**b**) HSPA5 spot 15 protein levels measured by 2D-DIGE and FVIII analysis (Spearman´s rank correlation coefficient). Abbreviations: 2D-DIGE—two-dimensional differential in-gel electrophoresis; LC—patient with lung cancer; C—healthy control.

**Figure 6 cancers-13-02260-f006:**
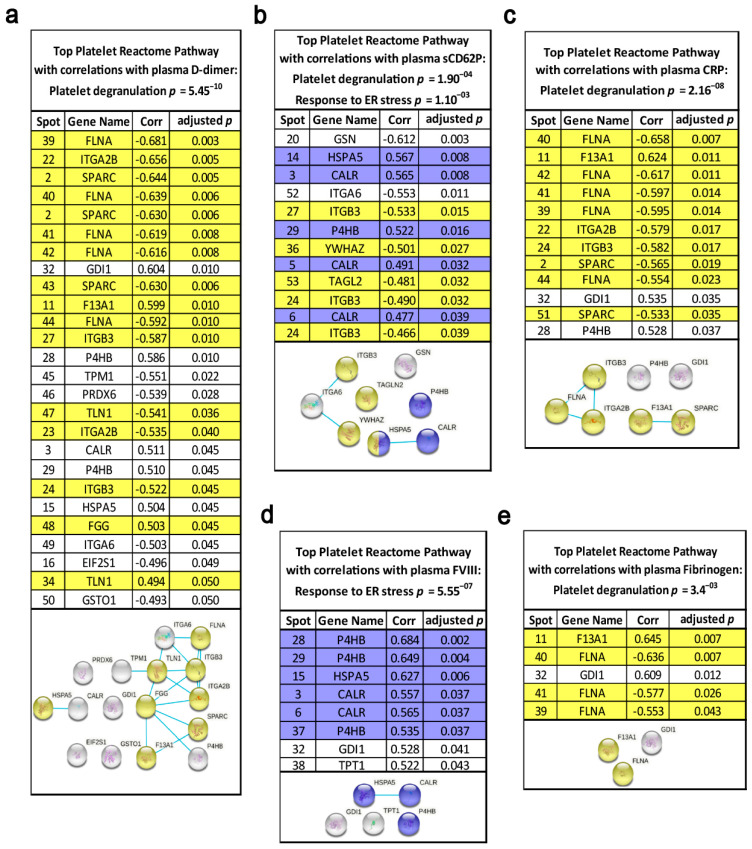
Functional relationships of the platelet proteome with haemostatic plasma laboratory parameters. The standardised abundance of the 566 included platelet protein spots from 2D-DIGE platelet proteome analysis were correlated with the plasma levels of (**a**) D-dimer (µg/mL), (**b**) sCD62P (ng/mL), (**c**) CRP (mg/dL), (**d**) FVIII activity (%) and (**e**) fibrinogen (mg/dL) by Spearman‘s rank correlation coefficient, and corresponding adjusted *p*-values are specified. These calculations were made from patients with lung cancer (*n* = 19) and matched controls (*n* = 19). Multiple comparisons were corrected by Benjamini–Hochberg. Network and enrichment analysis shows top pathways obtained upon entering the respective set of significantly correlating platelet proteins into the STRING database analysis tool. Known interactions are indicated by linear slope from curated databases. The proteins highlighted in grey in the STRING graphs were not significantly associated with these quoted functional networks.

**Figure 7 cancers-13-02260-f007:**
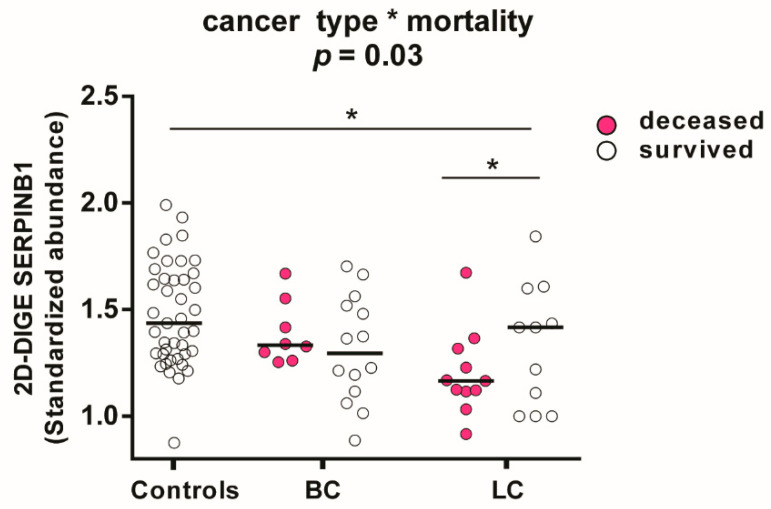
Influence of cancer type and mortality on SERPINB1 abundance in platelets measured by 2D-DIGE. Protein levels of SERPINB1 (leukocyte elastase inhibitor) are specified as standardized abundance. All values are depicted as median. The influence of cancer type and mortality was tested by a two-way ANOVA with the interaction term “cancer type*mortality” from patients with lung cancer (*n* = 11 non-survivors; *n* = 8 survivors) and brain cancer (*n* = 8 non-survivors; *n* = 14 survivors) and all matched controls (*n* = 41). The significance level is set to *p* < 0.05. * *p* < 0.05. Abbreviations: LC—patient with lung cancer; BC—patient with brain cancer.

**Table 1 cancers-13-02260-t001:** Baseline demographic, clinical and laboratory data of the study cohorts.

Characteristic	Healthy Controls (*n =* 41)	Healthy Controls for Brain Cancer (*n =* 22)	Healthy Controls for Lung Cancer (*n =* 19)	Brain Cancer (*n =* 22)	Lung Cancer (*n =* 19)
Median age at study entry, y (IQR)	57 (48–62)	55 (42–60)	61 (56–63)	56 (45–63)	62 (58–67)
Female, *n* (%)	16 (39)	9 (41)	7 (37)	9 (41)	7 (37)
**Median laboratory values (IQR)**					
Leukocyte (G/L)	5.9 (5.1–7.1)	5.7 (4.9–7.8)	5.9 (5.2–6.9) ^$$$^	7.1 (5.4–11.8)	9.8 (7.6–11.1)
Hemoglobin (G/dL)	14.5 (13.2–15.4)	14.8 (13.1–15.6)	14.3 (13–15.3) ^$^	14.3 (13.3–15.2)	12.9 (12.2–14)
Platelet count (G/L)	235 (216–286)	235 (214.5–308.5)	230 (225–286) ^$^	239 (202–297)	297 (276–350)
MPV (fl)	10.8 (10–11.4)	10.9 (10.1–11.4) ^#^	10.8 (9.9–11.5) ^$^	10 (9.7–10.7)	10.1 (9.2–10.8)
Neutrophils (%)	58 (51–64.5)	60 (48–61.8) ^###^	53 (54–68) ^$$$^	73 (56.7–76.1)	82.4 (76–87.7)
CRP (mg/dL)	0.14 (0.06–0.25)	0.16 (0.06–0.23)	0.09 (0.07–0.32) ^$$$^	0.14 (0.06–0.27)	1.16 (0.65–2.8)
aPTT (s)	35 (32.1–37.7)	33 (33.1–38.1) ^#^	36 (31.3–35.8)	34.1 (29.4–36)	33.5 (31.8–40.7)
Fibrinogen (mg/dL)	303 (259–337)	313 (250.5–319. 5) ^##^	282 (280–353) ^$$$^	319 (305–375)	509 (427–639)
Prothrombinfragment (pmol/L)	180 (127–249)	192 (118.5–297)	165 (148.5–225.8)	165 (109–199)	235 (154–349)
FVIII activity (%)	147 (107–173)	165 (103.5–154) ^###^	135 (117–184) ^$$^	208 (167–265)	237 (166–369)
Antithrombin III (%)	103 (97–108)	105 (96.5–108.5) ^###^	100 (101–108)	122 (111–133)	105 (85–110)
PAI (IU/mL)	1.2 (0.5–5)	1.1 (0.49–6)	1.5 (0.49–4.2)	2.1 (0.53–5.7)	1.4 (0.9–6.6)
Plasma FXIII activity (%)	123 (111–140)	122.5 (104.5–135.4) ^#^	120.4 (115.5–151)	106.8 (91.1–111)	120.2 (95.6–139.4)
D-dimer (µg/mL)	0.32 (0.27–0.42)	0.33 (0.27–0.41) ^#^	0.3 (0.27–0.42) ^$$$^	0.74 (0.33–0.93)	1.79 (1.13–4.15)
**VTE during follow up, ** ***n* (%)**	n.a.	n.a	n.a	2 (9.1)	2 (10.5)
PE	n.a	n.a	n.a	1 (5)	2 (10.5)
DVT	n.a	n.a	n.a	1 (5)	n.a.
**Deaths during follow up,** ***n* (%)**	n.a	n.a	n.a	8 (36.4)	11 (57.9)

The null hypothesis between two study groups was tested by Mann–Whitney U test. Statistically different values between groups: ^#^
*p <* 0.05, ^##^
*p <* 0.01, ^###^
*p <* 0.001 brain healthy controls vs. patients with brain cancer. ^$^
*p <* 0.05, ^$$^
*p <* 0.01, ^$$$^
*p <* 0.001 lung healthy controls vs. patients with lung cancer. Abbreviations: y—years; n—number; IQR—interquartile range; CRP—c-reactive protein; aPTT—activated partial thromboplastin time; FVIII—coagulation factor VIII; FXIII—coagulation factor XIII; PAI—plasminogen-activator-inhibitor; MPV—mean platelet volume; VTE—venous thromboembolism; PE—pulmonary embolism; DVT—deep vein thrombosis, n.a.—not applicable.

**Table 2 cancers-13-02260-t002:** 2D-DIGE-identified proteome alterations in platelets from patients with brain and lung cancer compared to healthy controls.

						Brain Cancer Patients/ Healthy Controls			Lung Cancer Patients/ Healthy Controls		
Spot Number	Protein Name	Gene Name	Isoelectric Point (pI)	MW (Da)	One-Way ANOVA (Adjusted)	Average FC	*p*-Value (Unadjusted)	*p*-Value (Adjusted)	Average FC	*p*-Value (Unadjusted)	*p*-Value (Adjusted)
1	Albumin	ALB	6.00	69,367	0.0224	0.84	0.0114	0.1136	**0.85**	**0.0281**	**0.0388**
2	Basement-membrane protein 40 (SPARC; Osteonectin)	SPARC	4.75	42,014	0.0488	0.96	0.6911	0.8639	**0.62**	**0.0001**	**0.0013**
3	Calreticulin	CALR	4.29	67,014	0.0127	1.06	0.1427	0.4926	**1.10**	**0.0315**	**0.0420**
4			4.29	65,300	0.0224	1.11	0.0254	0.1451	1.12	0.0524	0.0655
5			4.29	64,385	0.0264	1.09	0.0301	0.1505	1.11	0.0616	0.0747
6			4.29	63,493	0.0188	1.13	0.0069	0.1053	**1.11**	**0.0156**	**0.0240**
7			4.29	61,919	0.0433	1.14	0.0191	0.1273	**1.12**	**0.0123**	**0.0223**
8			4.29	60,228	0.0482	1.18	0.0074	0.1053	1.16	0.0515	0.0655
9			4.29	56,324	0.0224	1.14	0.0079	0.1053	**1.12**	**0.0054**	**0.0135**
10			4.29	57,598	0.0500	1.19	0.0142	0.1136	**1.12**	**0.0144**	**0.0238**
11	Coagulation factor XIII A chain	F13A1	5.00	55,000	0.0025	0.95	0.4346	0.7243	**1.57**	**0.0016**	**0.0080**
12			5.75	83,267	0.0500	0.90	0.2608	0.5611	**1.21**	**0.0102**	**0.0194**
13	Endoplasmic reticulum resident protein 29	ERP29	5.95	27,022	0.0188	1.06	0.2242	0.501	**1.20**	**0.013**	**0.0212**
14	Endoplasmic reticulum chaperone BiP	HSPA5	5.06	73,539	0.0488	0.99	0.8875	0.9342	1.07	0.1212	0.1347
15			5.08	73,539	0.0224	1.06	0.3828	0.6960	**1.15**	**0.0098**	**0.0194**
16	Eukaryotic translation initiation factor 2 subunit 1	EIF2S1	5.08	43,211	0.0188	0.99	0.8726	0.9342	**0.83**	**0.0018**	**0.0080**
17	Fermitin family homolog 3	FERMT3	6.00	39,072	0.0019	0.94	0.5676	0.8148	1.11	0.1191	0.1347
18	Fibrinogen gamma chain	FGG	5.74	55,146	0.0467	1.03	0.4800	0.7385	**1.18**	**0.0022**	**0.0080**
19			5.58	56,048	0.0488	1.45	0.2812	0.5624	**1.47**	**0.0032**	**0.0091**
20	Gelsolin	GSN	5.70	81,086	0.0188	1.07	0.0989	0.3956	**0.92**	**0.0221**	**0.0316**
21	Integrin alpha-IIb	ITGA2B	4.80	88,938	0.0467	0.94	0.1933	0.5155	**0.88**	**0.0052**	**0.0135**
22			4.92	88,938	0.0096	0.93	0.1562	0.4926	**0.86**	**0.0001**	**0.0013**
23			4.95	88,938	0.0224	0.91	0.0641	0.2849	**0.86**	**0.0020**	**0.0080**
24	Integrin beta-3	ITGB3	4.68	75,464	0.0127	1.00	0.9763	0.9763	**0.82**	**0.0008**	**0.0053**
25			4.70	75,464	0.0390	0.95	0.2597	0.5611	**0.87**	**0.0061**	**0.0144**
26			4.73	75,464	0.0467	0.96	0.3805	0.6960	**0.88**	**0.0094**	**0.0194**
27			5.63	75,464	0.0046	0.90	0.2665	0.5611	**0.63**	**0.0005**	**0.0040**
28	Protein disulfide-isomerease	P4HB	4.78	61,026	0.0127	1.01	0.8415	0.9342	**1.11**	**0.0016**	**0.0080**
29			4.80	61,026	0.0019	0.98	0.5906	0.8148	**1.11**	**0.0211**	**0.0313**
30	Protein disulfide-isomerase A3	PDIA3	5.81	60,726	0.0467	1.02	0.4246	0.7243	1.03	0.2144	0.2199
31			5.81	58,443	0.0417	1.04	0.1645	0.4926	1.06	0.1529	0.1653
32	Rab GDP dissociation inhibitor alpha	GDI1	4.95	62,247	0.0019	1.02	0.6780	0.8639	**1.31**	**0.0002**	**0.0020**
33	Serotransferrin	TF	6.62	76,915	0.0272	0.95	0.4797	0.7385	**0.79**	**0.0149**	**0.0238**
34	Talin-1	TLN1	5.31	75,981	0.0096	0.97	0.7319	0.8872	**1.45**	**0.0072**	**0.0160**
35	14-3-3 protein epsilon	YWHAE	4.60	28,953	0.0420	1.01	0.5907	0.8148	0.98	0.3459	0.3459
36	14-3-3 protein zeta/delta	YWHAZ	4.69	26,607	0.0420	0.99	0.8331	0.9342	0.96	0.0673	0.0792

The one -way ANOVA *p*-values indicate the variance of the particular proteoforms between patients with brain and lung cancer and matched healthy controls. The average fold change of 2D-DIGE-quantified protein spot abundance and *p*-values for unadjusted and adjusted post-hoc contrast testing are calculated between the patients with lung cancer (*n* = 19) and brain cancer (*n* = 22) and the respective matched healthy controls (altogether *n* = 41). Statistically significant correlations are highlighted in bold.

**Table 3 cancers-13-02260-t003:** F13A1 proteoform abundance and their respective correlation with enzymatic F13A1 activities in platelets. The one-way ANOVA *p*-values indicate the variance of the particular F13A1 proteoform between patients with brain and lung cancer and matched healthy controls. The average fold change of 2D-DIGE-quantified F13A1 spot abundances and *p*-values for explorative post-hoc contrast testing are calculated between the patients with lung cancer (*n* = 19) and brain cancer (*n* = 22) and the respective matched healthy controls (altogether *n* = 41). The correlations of each F13A1 proteoform abundance with the F13A1 enzymatic activity of the respective platelet samples were exploratorily assessed by Spearman’s rank correlation coefficient and corresponding *p*-values from all patients with brain and lung cancer and matched controls.

				Lung Cancer Patients/ Matched Controls		Brain Cancer Patients/ Matched Controls		Correlation: Abundance vs. Enzymatic Activity	
Spot Number of F13A1 Proteoforms	Isoelectric Point (pI)	MW (kDa)	ANOVA (Adjusted)	Average FC	*p*-Value (Unadjusted)	Average FC	*p*-Value (Unadjusted)	corr. of F13A1 Abundance vs. Activity→Rho	*p*-Value (Unadjusted)
12c	5.85	83	0.998	0.77	0.2652	1.24	0.372	−0.29	0.062
12b	5.75	83	0.410	1.40	0.0019	0.96	0.743	0.43	0.004
12	5.60	83	0.071	1.21	0.0102	0.90	0.261	0.39	0.009
12a	5.65	83	0.198	1.33	0.0369	0.82	0.231	0.50	0.001
12d	6.05	79	0.868	1.26	0.3055	1.05	0.832	0.04	0.807
11	4.95	55	0.006	1.57	0.0016	0.95	0.433	0.14	0.367

## Data Availability

The data presented in this study are partially available in the Appendix A. All data are available on reasonable request from the corresponding author.

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
