# Peer review of "Alterations of the Platelet Proteome in Lung Cancer: Accelerated F13A1 and ER Processing as New Actors in Hypercoagulability"

_cancers, 2021, doi:10.3390/cancers13092260_

Round 1

Reviewer 1 Report

All my concerns from the previous submission have been adequately addressed in the revised manuscript. The authors have performed additional experiments, which support their conclusions.

Reviewer 2 Report

Authors in R1 version, have revised my remarks. Thus I consider to publish manuscript at current form

This manuscript is a resubmission of an earlier submission. The following is a list of the peer review reports and author responses from that submission.

Round 1

Reviewer 1 Report

In the manuscripts by Ercan et al, the authors investigated changes in the platelet proteome of brain and lung cancer patients. These kinds of malignancies have an increased risk of developing VTE, but the mechanism of thrombosis is still not understood. Although the authors identified changes in several platelet proteins from lung cancer patients, the validation for some of these proteins are missing. The manuscripts is written in a much disorganized way, and a lot of redundancies are found within the results section. More efforts should be done to organize the results and figures. 

Major comments

  1. In the 3.1 section of the manuscript, the authors did not comment on the statistical differences found in some coagulation markets, they should describe these differences here.
  2. Section 3.2 is too long, there are paragraphs that can be summarized and information that is redundant and can be cut or moved to the method section. For example the main aim in page 10 is to describe the results showed in table 2, the most relevant information is in the sentences 363-366 and sentences 375-378. The description of the statistical analysis shouldn’t be mentioned again in the results section, it is already in the methods section and in the figures and table legends. The paragraph describing that no significant correlation was detected between platelets count and protein spots can be summarized in a couple of sentences and it should be put in the method section as a control evaluation. All the important information is in table 2, figure 2 is redundant, it does not add more information and it can be removed or put as a supplemental figure. Also Sentence 378 is unfinished.
  3. It would be easy to interpretative the results if the proteins full name were mentioned in the text too.
  4. Section 3.3 can be summarized too. The description of the NetworkAnalyst approach is in the method section, so it should not be in the results again. Also figure 4 can be a supplemental figure, it does not add more information than the results showed in figure 3. The authors decided to validate F13A1, P4HB, CALR and HSPA5 but the validation of P4HB is missing. Why did the authors not study P4HB in the platelets samples? Why were P4HB levels not measured in plasma? Moreover, the authors choose not to validate ITGA2B, ITGB3 and TLN1, but they showed a representative WB image of the platelet proteome in a supplementary figure (figure S5). Why the authors put these results in a supplementary figure when the differences in the levels of these proteins seems to be significant?
  5. The results showed in section 3.4 of the manuscript should be reorganized. The authors jumped from one figure to another when describing this result section and it makes very difficult to interpret the results. Figure 5a and figure S1 are the same. Figure 5a should be removed. Figure 5b-c are not mentioned in the text and these results are in table 3, so the authors should decide whether to show the data from in a graph or in a table. The information in table S1 is very confusing and does not clarify the results obtained, it should be removed. More focus should be done on the interesting finding that the 55 kDa F13A1 fragment comes from the cleavages of the full-length factor XIII. The authors should have talked about this finding much earlier in this result section, followed by the results in figure 5d-f. Also, the authors explained the result in figure 5f before talking about panel 5d and 5e. These figures should be reorganized.
  6. The authors hypothesize that plasmin generates the 55 kDa F13A1 fragment, they should measure fibrinolytic markers in plasma such as plasmin-α2-antiplasmin complexes and see if lung cancer patients have increased plasmin-α2-antiplasmin complexes levels.
  7. Figure 6b and 6c are not mentioned in the text. The authors identified three more CALR and two more HSPA5 spots, which they seem to be significant. Why in this case did the authors not add a table with these data?
  8. Table S2 is not mentioned in the text
  9. The authors identified decreased neutrophil elastase levels in deceased patients compared to surviving patients. Did the authors measure plasma H3Cit levels as marker of NETosis?
  10. Why did the authors not talk about the elastase result in the discussion section? Instead, the authors dedicate one paragraph to discuss the result of a supplementary figure (Figure S5). I do not understand why the authors decided not to focus on these results, which are significant, but talk about them in the discussion.

Reviewer 2 Report

The paper presents a comparative analysis to study protein panel in the platelet of two subjects. These are the group of individuals that were prone to thrombosis (patients suffering with brain or lung cancer) and healthy control group serving to confront the results.

As the result, in lung cancer patients, the altered ER and other organelle proteomes were confirmed.

As the ultimate outcome, new targets in platelets such as chaperones etc. were assumed to be therapeutic targets in the lung cancer.

REMARKS

INTRODUCTION (3rd paragraph, note)

Unfortunately, you´ve jumped directly to describe only the proteomics based on F-2D-DIGE referring to analyse cancer-related changes in protein down- up-regulations and PTMs. Here, I miss strongly the general description in-brief, of the current status of proteomics enabling to get comprehensive and a large-scale datasets related also to the latest efforts to make the very high-throughput proteomic schemes.

INTRODUCTION (3rd paragraph, upgrade)

The state-of-the-art proteomic technologies enable currently to analyze thousands of proteins while the very high-throughputs can be achieved [https://doi.org/10.1021/acs.analchem.0c00752]. Here, liquid-handling workstations are the main example [https://doi.org/10.1016/j.cca.2020.04.015]. The modern proteomics approach such as fluorescence-based two-dimensional differential gel …

MATERIALS AND METHODS, RESULTS, CONCLUSION

These parts are exhaustive in providing relevant data of acquired protein panel, involved in the molecular pathways analyses. The length of explanations is fairly acceptable and comprehensive.

What I miss is to draw a procedure workflow to describe easily what has been done and to show quickly to the reader the procedure pipeline for its fast understanding.

In the conclusion, write your future plans in this field of studying.

Author Response

We would like to thank all of the reviewers and the guest editor for giving us the opportunity and helpful suggestions to undertake a major revision for the publication of our work in “Cancers”. In the following section “reviewer’s comments” (in bold), answers and new data to the individual comments and questions of the reviewers are given point by point. In the manuscript main changes are highlighted in blue.

Reviewer 2:

The paper presents a comparative analysis to study protein panel in the platelet of two subjects. These are the group of individuals that were prone to thrombosis (patients suffering with brain or lung cancer) and healthy control group serving to confront the results.

As the result, in lung cancer patients, the altered ER and other organelle proteomes were confirmed.

As the ultimate outcome, new targets in platelets such as chaperones etc. were assumed to be therapeutic targets in the lung cancer.

REMARKS

INTRODUCTION (3rd paragraph, note)

Unfortunately, you´ve jumped directly to describe only the proteomics based on F-2D-DIGE referring to analyse cancer-related changes in protein down- up-regulations and PTMs. Here, I miss strongly the general description in-brief, of the current status of proteomics enabling to get comprehensive and a large-scale datasets related also to the latest efforts to make the very high-throughput proteomic schemes.

INTRODUCTION (3rd paragraph, upgrade)

The state-of-the-art proteomic technologies enable currently to analyze thousands of proteins while the very high-throughputs can be achieved [https://doi.org/10.1021/acs.analchem.0c00752]. Here, liquid-handling workstations are the main example [https://doi.org/10.1016/j.cca.2020.04.015]. The modern proteomics approach such as fluorescence-based two-dimensional differential gel …

Thank you for the helpful suggestion and support in mentioning the status of the current proteomics technology in the introduction. We have included the proposed paragraph and reference in the introduction (page: 2-3; third paragraph).

MATERIALS AND METHODS, RESULTS, CONCLUSION

These parts are exhaustive in providing relevant data of acquired protein panel, involved in the molecular pathways analyses. The length of explanations is fairly acceptable and comprehensive.

What I miss is to draw a procedure workflow to describe easily what has been done and to show quickly to the reader the procedure pipeline for its fast understanding.

Thank you, we have now added an illustration (Figure 1) of the detailed study process in addition to the graphic summary (page: 9).

In the conclusion, write your future plans in this field of studying.

Thank you for this suggestion. We described some of our future plans in the conclusion (page: 23).